# Impact of the 21-Gene Assay in Patients with High-Clinical Risk ER-Positive and HER2-Negative Early Breast Cancer: Results of the KARMA Dx Study

**DOI:** 10.3390/cancers15051529

**Published:** 2023-02-28

**Authors:** Antonio Llombart-Cussac, Antonio Anton-Torres, Beatriz Rojas, Raquel Andrés, Noelia Martinez, César A. Rodríguez, Sara Marin, Teresa Puértolas, Alejandro Falcón González, María Leonor Fernández-Murga, Carlos Hagen, Manuel Ruiz-Borrego

**Affiliations:** 1Oncology Department, Hospital Arnau de Vilanova, Fundación para el Fomento de la Investigación Sanitaria y Biomédica de la Comunitat Valenciana (FISABIO), 46020 Valencia, Spain; 2Oncology Department, Universidad Católica, 46900 Valencia, Spain; 3Oncology Department, Hospital Miguel Servet, 50009 Zaragoza, Spain; 4Oncology Department, Hospital Vall d’Hebron, VHIO, 08035 Barcelona, Spain; 5Oncology Department, Hospital Lozano Blesa, 50009 Zaragoza, Spain; 6Oncology Department, Hospital Universitario Ramon y Cajal, 28034 Madrid, Spain; 7Oncology Department, Hospital Universitario de Salamanca, IBSAL, 37007 Salamanca, Spain; 8Oncology Department, Hospital Universitario Virgen del Rocio, 41013 Sevilla, Spain; 9Oncology Division, Palex Medical SA, 28108 Madrid, Spain

**Keywords:** adjuvant, breast cancer, chemotherapy, clinical utility, Oncotype DX Breast Recurrence Score^®^, Recurrence Score^®^ result

## Abstract

**Simple Summary:**

The 21-gene Oncotype DX Breast Recurrence Score^®^ assay is prognostic and predictive of chemotherapy benefit for patients with estrogen receptor-positive, human epidermal growth factor receptor 2-negative (HER2−) in Early Breast Cancer (EBC). The KARMA Dx study evaluated the impact of the Recurrence Score^®^ results (RS) on the treatment decision for patients with EBC and high-risk clinicopathological characteristics for whom chemotherapy (CT) was considered. A total of 219 consecutive patients were included. After 21-gene testing, treatment decisions changed from CT + endocrine therapy (ET) to ET alone for 67% of the whole group. Physicians’ confidence in their final recommendations increased in 34% of cases. Our findings indicate the substantial potential of the 21-gene test to guide CT recommendations in patients with EBC considered to be at high risk of recurrence based on clinicopathological parameters, regardless of nodal status or treatment setting.

**Abstract:**

Background: The 21-gene Oncotype DX Breast Recurrence Score^®^ assay is prognostic and predictive of chemotherapy benefit for patients with estrogen receptor-positive, HER2− early breast cancer (EBC). The KARMA Dx study evaluated the impact of the Recurrence Score^®^ results (RS) on the treatment decision for patients with EBC and high-risk clinicopathological characteristics for whom chemotherapy (CT) was considered. Methods: Eligible patients with EBC were candidates for the study if CT was considered standard recommendation by local guidelines. Three high-risk EBC cohorts were predefined: (A) pT1-2, pN0/N1mi, and grade 3; (B) pT1-2, pN1, and grades 1–2; and (C) neoadjuvant cT2-3, cN0, and Ki67 ≤ 30%. Treatment recommendations before and after 21-gene testing were registered, as well as treatment received and physicians’ confidence levels in their final recommendations. Results: A total of 219 consecutive patients were included from eight Spanish centers: 30 in cohort A, 158 in cohort B, and 31 in cohort C. Ten patients were excluded from the final analysis as CT was not initially recommended. After 21-gene testing, treatment decisions changed from CT + endocrine therapy (ET) to ET alone for 67% of the whole group. In total, 30% (95% confidence interval [CI] 15% to 49%), 73% (95% CI 65% to 80%), and 76% (95% CI 56% to 90%) of patients ultimately received ET alone in cohorts A, B, and C, respectively. Physicians’ confidence in their final recommendations increased in 34% of cases. Conclusions: Use of the 21-gene test resulted in an overall 67% reduction in CT recommendation in patients considered candidates for CT. Our findings indicate the substantial potential of the 21-gene test to guide CT recommendations in patients with EBC considered to be at high risk of recurrence based on clinicopathological parameters, regardless of nodal status or treatment setting.

## 1. Introduction

Defining the best therapeutic strategy and optimal systemic therapy in patients with endocrine receptor-positive (ER+) human epidermal growth factor receptor 2-negative (HER2−) early breast cancer (EBC) is often a complex decision. Clinicopathological features, including histologic grade, tumor size and presence of axillary lymph-node metastases, provide limited prognostic information, however, any of them have shown to be predictive of benefit from chemotherapy. Therefore, specific molecular signatures have been generated to provide personalized information on prognosis and to predict the likelihood of benefit from chemotherapy (CT) in ER+/HER2- EBC.

The predictive and prognostic value of the 21-gene assay (Oncotype DX Breast Recurrence Score^®^ assay; Exact Sciences Corp. formerly Genomic Health, Inc., Redwood City, CA, USA) in node-negative (N0) patients was confirmed in the prospective TAILORx study [1,2,3]. In patients with Recurrence Score^®^ results (RS) 11 to 25, the addition of CT to endocrine therapy (ET) showed no added benefit, with a 9-year overall survival of 93.9% and 93.8% for the ET and CT + ET arms, respectively [2]. Use of the 21-gene assay is supported by the highest level of evidence (1A) and incorporated in major international guidelines for treatment decisions in patients with N0, ER+, HER2− early breast cancer (EBC) [4,5,6,7]. A secondary analysis of the study confirms that the integration of clinical factors together with the RS provides greater precision in determining the risk of recurrence and guiding the use of systemic therapy, particularly among high-risk clinical patients [8]. The 21-gene test was also validated for its prognostic value and prediction of CT benefit for patients with node-positive (N1) disease [9,10]. The RxPONDER study has recently confirmed the lack of benefit of CT in postmenopausal patients with N1 (1–3 positive nodes) and RS 0–25 [11]. Finally, several neoadjuvant CT studies in ER+ HER2− EBC have shown consistent correlation between pathologically complete response rates and high RS results [12,13,14,15,16,17,18].

In Spain, national guidelines have incorporated molecular tests, particularly the 21-gene assay, for patients with ER+, HER2− EBC, and the 21-gene test has been confirmed as cost-effective for the national health system [19]. Still, patients with high-clinical risk EBC, defined as pN1, high tumor proliferation markers, or candidates for neoadjuvant CT are systematically excluded from reimbursement. We therefore assessed the impact of the 21-gene test on the treatment decisions in real-world practice for a cohort of patients with clinically high-risk ER+, HER2− EBC for whom CT was indicated without testing and clinicians had doubts about the real benefit of CT.

## 2. Methods

### 2.1. Study Design

This multicenter study evaluated the impact of the 21-gene Oncotype DX Breast Recurrence Score test on treatment recommendations made by physicians in Spain caring for patients with breast cancer. The study was carried out in accordance with Good Clinical Practice guidelines. Approval for the protocol was obtained from ethics committees at participating institutions. The 21-gene test was performed, and Recurrence Score^®^ results (RS) were calculated as described previously [20]. Physicians enrolled in the study completed a questionnaire at baseline indicating their initial treatment recommendations based on available clinical and pathological data. After 21-gene testing and discussion of test results, physicians completed separate follow-up questionnaires indicating whether the test results changed their initial treatment recommendations. Finally, for patients who were candidates for neoadjuvant therapy, treatments administered to patients over 6 months were documented. The choice of systemic adjuvant or neoadjuvant therapy was subject to physician discretion.

The primary objective was to determine whether the 21-gene test results affected physicians’ treatment recommendations by comparing recommendations made before and after 21-gene testing. The secondary objective was to determine whether the 21-gene test results affected physicians’ expressed level of confidence in their treatment recommendations.

### 2.2. Study Participants

Eligible physicians included medical oncologists who make adjuvant or neoadjuvant treatment recommendations to patients with breast cancer who were eligible for this study (see below). Participating physicians were those primarily responsible for explaining to patients the 21-gene test results and the rationale for their treatment recommendations.

Consecutive eligible patients were women 18–75 years of age, diagnosed with stage I or II, ER+ (defined as Allred score > 2), HER2− (defined as immunohistochemistry [IHC] 0 or 1+, or IHC 2+ and fluorescent/chromogenic in situ hybridization-negative) infiltrating breast cancer who were considered candidates for systemic CT. Patients represented, in the treating physician’s judgment, the population for which the benefit of systemic CT added to ET is either unclear or insufficient to warrant its use, despite clinical recommendations. Patients also fulfilled the eligibility criteria for one of three study cohorts: Cohort A (completed surgery within 6 weeks of study consent, T1c-3, lymph node-negative [pN0] or micrometastases [pN1mi], and tumor grade 3 [defined as Scarff-Bloom-Richardson (SBR) 8 or 9]); Cohort B (completed surgery within 6 weeks of study consent, 1–3 positive axillary lymph nodes confirmed by histology, and tumor grade 1/2 [defined as SBR < 8]); or Cohort C (scheduled for neoadjuvant CT followed by surgery, tumor size > 2 cm [T2 or T3, not T4], lymph node-negative by ultrasound, magnetic resonance imaging, or sentinel lymph node biopsy, and Ki67 ≤ 30% by local procedures). All patients provided signed informed consent.

Patients were ineligible if they were ER−, HER2+, diagnosed with more than one primary breast tumor, had multicentric tumors, had known metastatic breast cancer, or had a history of breast cancer in the same breast or of infiltrating breast cancer within the previous 5 years. Patients were also excluded if they had systemic ET within the previous year, or if they were not candidates for systemic CT.

### 2.3. Ethical Considerations

The local Ethics Committee of the Hospital Arnau de Vilanova approved this study: (HAV-BAR-18/2016). Legislation on data confidentiality was amended in 1999 (Organic Law 15/1999 on Data Protection BOE-A-1999-23750).

The authors have no relevant affiliations with any organization or entity with a financial interest in the subject matter or materials discussed in the manuscript. All enrolled patients in the study signed informed consent.

### 2.4. Statistical Analysis

Continuous variables were summarized using median and interquartile range. Categorical variables were summarized with frequencies and percentages. Pearson’s chi-square test was used to check for an association between change in treatment recommendation and RS groups. The Clopper-Pearson method was used to calculate 95% confidence intervals (CI) for the proportion of treatment recommendations that switched to no CT. In a post hoc analysis, the distribution of RS results using the cutpoints from the TAILORx trial [1,2,3] was done for N0 patients (Appendix A).

## 3. Results

### 3.1. Study Participants

Medical oncologists from 8 centers enrolled a total of 238 patients between July 2016 and November 2017. Nineteen patients were excluded for reasons of protocol violation (*n* = 13) and unresolved data query (*n* = 6). Of the remaining 219 patients, 30 (14%) were included in Cohort A, 158 (72%) in Cohort B, and 31 (14%) in Cohort C (Figure 1). Patient and disease characteristics are summarized in Table 1. Although patients had high-risk clinicopathologic features, namely high tumor grade, nodal involvement, or large tumor size, there was a wide distribution of RS results in each of the study cohorts (Figure 2).

### 3.2. Change in Treatment Recommendations after 21-Gene Testing

All patients were considered candidates for systemic CT based on clinicopathologic features; nonetheless, 10 (5%) patients—8 in Cohort B and 2 in Cohort C—had an initial recommendation for ET alone. The remaining analyses therefore included only the 209 patients who had initial recommendations for CT.

After RS results were available, physicians changed their treatment recommendations to omit CT for 67% of patients overall (95% CI 60% to 73%) (Figure 3). Specifically, physicians changed their recommendations for 30% of patients in Cohort A, 73% in Cohort B, and 76% in Cohort C (Figure 3).

Within each cohort, the change in treatment recommendations varied by RS group (Pearson chi-square *p* < 0.001), with the greatest reductions in CT recommendations observed for patients with RS 0–17 (Figure 4). Notably, 91% of patients in Cohort B, and 80% in Cohort C with a RS < 18 omitted CT.

Among the 23 patients in cohort C who did not receive neoadjuvant CT, 16 (70%) were also not recommended adjuvant CT. Of these, 14 (82%) had RS 0–17, and 2 (13%) had RS 18–30. All patients with RS 31–100 received CT.

### 3.3. Physician Confidence Level with 21-Gene Test Results

Physicians expressed higher or the same level of confidence in 90% of their post-test treatment recommendations for the overall cohort (*n* = 209) (Figure 5). In particular, 45% of physicians reported increased confidence in their post-test treatment recommendations for patients in Cohort C. Additionally, 73% of physicians expressed unchanged confidence in their post-test treatment recommendations for patients in Cohort A.

## 4. Discussion

The KARMA Dx study aimed to define the utility of the 21-gene test in three cohorts of patients with intermediate- to high-risk clinical or pathological characteristics that excluded them from local access to the 21-gene test and for whom local or national guidelines were not conclusive regarding the use of (neo)adjuvant CT. Our results show that granting access to the 21-gene testing led to clinicians to reject CT in 67% of patients with well-defined intermediate- to high-risk clinicopathological ER+, HER2− EBC.

Several studies had explored the treatment decision changes based on the 21-gene test. In a Spanish study limited to N0 patients, we reported a change in treatment recommendations in 32% of patients [21]. The French PONDx study, including 882 patients with N0 or N1 disease, reported an absolute reduction in CT recommendations of 36% [22]. With a similar design, the Italian PONDx study analyzed the changes in treatment decisions in a cohort of 1738 patients. RS result-guided treatment decisions resulted in a 36% reduction of CT recommendations [23]. The CT de-escalation rates in all those studies were lower than that observed in the KARMA Dx study and may reflect differences in study design and patient selection. Each of the three cohorts in our study reflects a particular scenario in which clinicians demand reliable and trustworthy tools to better individualize risk and benefits from CT, even though a recommendation for CT is reasonable.

Cohort A (N0/N1mi, and grade 3) was the smallest, comprising only 14% of all patients tested in the study. In a pooled analysis of 4 European studies on the 21-gene testing including 565 patients with N0 EBC, only 13% had grade 3 tumors, and the global treatment de-escalation was modest, as CT was maintained in 70% to 73% of patients [24]. Similarly, in our study, cohort A achieved the lowest rate of CT de-escalation, with only 30% of patients avoiding CT. This rate was substantially lower than those of the other two cohorts (73% and 75% for cohorts B and C, respectively). Cohort A had also the greatest proportion of patients with RS results 31–100 (33% vs. 4% and 13% for cohorts B and C, respectively). Tumor grade is prognostic and independently associated with risk for recurrence; however, histopathologic high grade does not correlate well with the risk provided by the RS result. In the PlanB trial, approximately 50% of grade 3 tumors had an RS result less than 31 [25]. A national cohort analysis including 13,558 women with N0 and grade 3 tumors showed that 27.1% had an RS result less than 18. The five-year overall survival rate for patients with a low RS result was not significantly higher with CT (absolute differences: 2.5%, *p* =0.07) [25]. Cohort A had the highest proportion of RS results 0–17 for whom physicians maintained the CT recommendations (45%; *n* = 4/9), which reflects that many women with grade 3 continue to be offered CT irrespective of RS results. Finally, our results confirm the value of the 21-gene test in patients with N0/N1mi and biologically aggressive characteristics.

Cohort B (N1/grade 1–2) comprised 72% of patients and achieved a 73% de-escalation in CT after 21-gene testing. The final CT recommendation strongly correlated with RS results: 9% (8/86) with RS results 0–17, 46% (26/57) with RS results 18–30, and 100% (7/7) with RS results 31–100. Before the final results from prospective RxPONDER study, accruing evidence had emerged from retrospective and real-world studies supporting the 21-gene test as a reliable prognostic and potentially predictive tool for chemotherapy in N1 patients [9,10,26,27,28,29,30]. The retrospective-prospective analysis from the SWOG S8814 trial found no benefit from CT for patients with RS results < 18, while demonstrating substantial benefit for those with RS results > 30, and an uncertain benefit for those with RS results 18–30 [10]. The prospective West German Study Group (WSG) PlanB study treated 338 patients with N0 or N1 and RS results < 11 with ET alone. The five-year disease-free survival was 94.4% for the N1 group and 94.2% for the N0 group [28].

In RxPONDER trial, 5018 patients with N1 (1 to 3) ER+/HER2− EBC and RS results 0–25 were randomly assigned to receive CT followed by ET or ET alone. With a median follow-up of 5.1 years, the trial concluded that there was no significant difference in invasive disease-free survival (iDFS) in the intention-to-treat population or among postmenopausal patients. However, among premenopausal women, invasive disease–free survival at 5 years was 89.0% with endocrine-only therapy and 93.9% with CT followed by ET (hazard ratio, 0.60; 95% CI, 0.43 to 0.83; *p* = 0.002) [11]. This finding is in line with TAILORx reporting benefits for CT in patients ≤ 50 years of age with RS results 21–25 [2]. It has been suggested that the benefits from CT seen in the premenopausal groups may be due to its impact on ovarian function. At the time both trials were recruiting, ovarian function suppression was not yet a standard approach for high-risk premenopausal patients [31].

Cohort C included patients with large tumors (>3 cm), low to moderate Ki67 (≤30%), and absence of axillary involvement. This cohort included a younger population with a large tumor size. Very few trials have explored this patient population. Nevertheless, results and confidence of physicians were in line with the other two cohorts. Up to 75% of patients had a final decision change, suggesting the high uncertainty of clinicians to recommend neoadjuvant CT in this population. The TransNEOS study validated the 21-gene test as a predictor of clinical response to neoadjuvant letrozole in postmenopausal patients [32]. Ongoing studies are evaluating the role of the 21-gene test in this setting. In the ADAPT study (NCT01779206), patients with RS results < 25 and early sensitivity to ET are guided to ET, while the other patients are randomized to different CT regimens. In a different approach, the DxCartes study (NCT03819010) is evaluating whether neoadjuvant letrozole and palbociclib can induce long-term changes in RS results in those with baseline RS results > 18.

This study was designed prior to the new RS score cut-off recommendations that emerged from the TAILORx study for node-negative patients. In this study, a cut-off point of the RS was established for the benefit of CT of >25 for women > 50 years of age and >16 for younger women [2]. These results were published while the study was in progress, and it is inevitable to consider that they were incorporated into the decision-making process of the researchers. We have performed an unplanned analysis for cohorts A and C in the final CT decision (Appendix A). Decisions to omit CT based on the new guidelines are even more consistent with the RS result, and particularly for postmenopausal patients.

Our study presents several limitations. Cohorts A and C were under-represented. The study was designed and conducted long before availability of TAILORx and RxPONDER results. It is likely that the updated RS result classification, and particularly the uncertainty of CT benefits in premenopausal patients with low RS results will have had an impact on the decision process. The study captures the complete treatment received up to 6 months after RS results were available. We cannot rule out that patients in the neoadjuvant cohort who were spared neoadjuvant CT did not receive adjuvant CT following neoadjuvant ET and surgery. Finally, clinical outcomes were not collected, although the study was not powered for this analysis.

The role of CT in ER+, HER2− Early Breast Cancer (EBC) has been redefined with the introduction of the 21-gene test. The prospective TAILORx and RxPONDER studies have provided level 1A evidence of the effectiveness of the 21-gene test to guide CT recommendations in HR+, HER2− EBC patients. Our study shows that incorporating the 21-gene test in predefined intermediate to high-risk groups, for which CT was recommended by national guidelines, results in a substantial change in the final decision favoring CT de-escalation. Use of the 21-gene test in the N1 and neoadjuvant scenarios under reasonable pathological criteria have great impact on physicians’ decision to spare CT when it is not shown to add clinical benefit.

## 5. Conclusions

Oncotype DX Breast Recurrence Score testing resulted in an overall 67% reduction in CT recommendations in patients considered candidates for CT based on high-risk clinicopathologic criteria. Our findings highlight the potential of the Oncotype DX^®^ test to guide physicians’ decisions on CT recommendations with confidence, particularly in patients considered to be at clinical high risk, regardless of treatment setting (adjuvant or neoadjuvant)

## Figures and Tables

**Figure 1 cancers-15-01529-f001:**
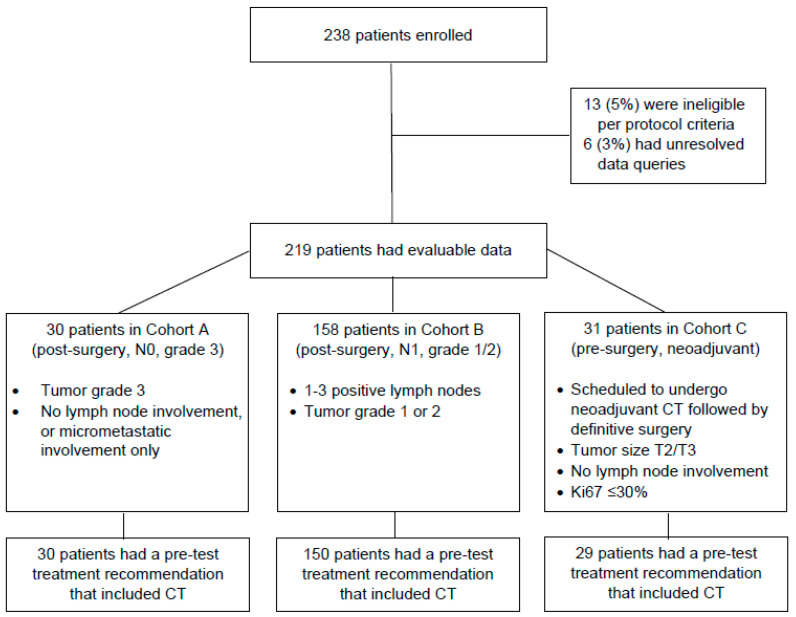
Study flow diagram. CT, chemotherapy; N0, node-negative; N1, node-positive.

**Figure 2 cancers-15-01529-f002:**
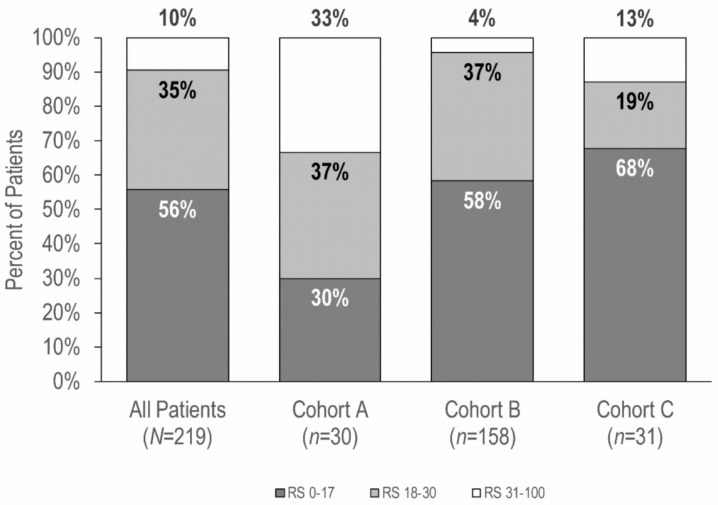
Distribution of Recurrence Score groups, by cohort (*n* = 219). Percentages may not sum to 100% because of rounding. RS, Recurrence Score result.

**Figure 3 cancers-15-01529-f003:**
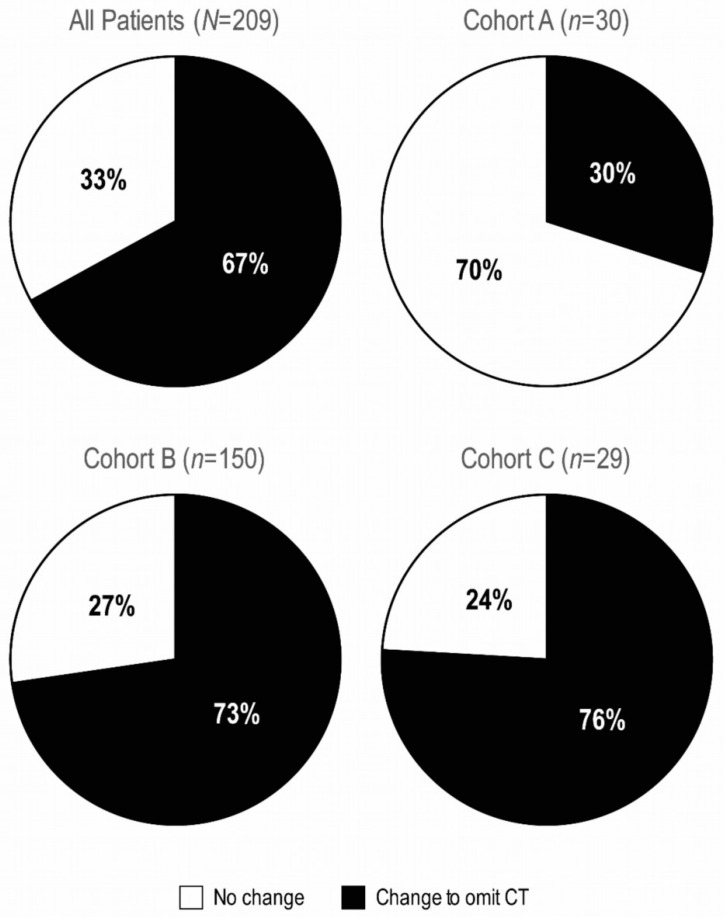
Change in treatment recommendations after 21-gene testing by cohort (*n* = 209). Change to omit CT indicates a change from CT + ET → HT in Cohorts A and B, and from CT → ET in Cohort C. CT, chemotherapy; CT + ET, chemotherapy plus endocrine therapy; ET, endocrine therapy alone.

**Figure 4 cancers-15-01529-f004:**
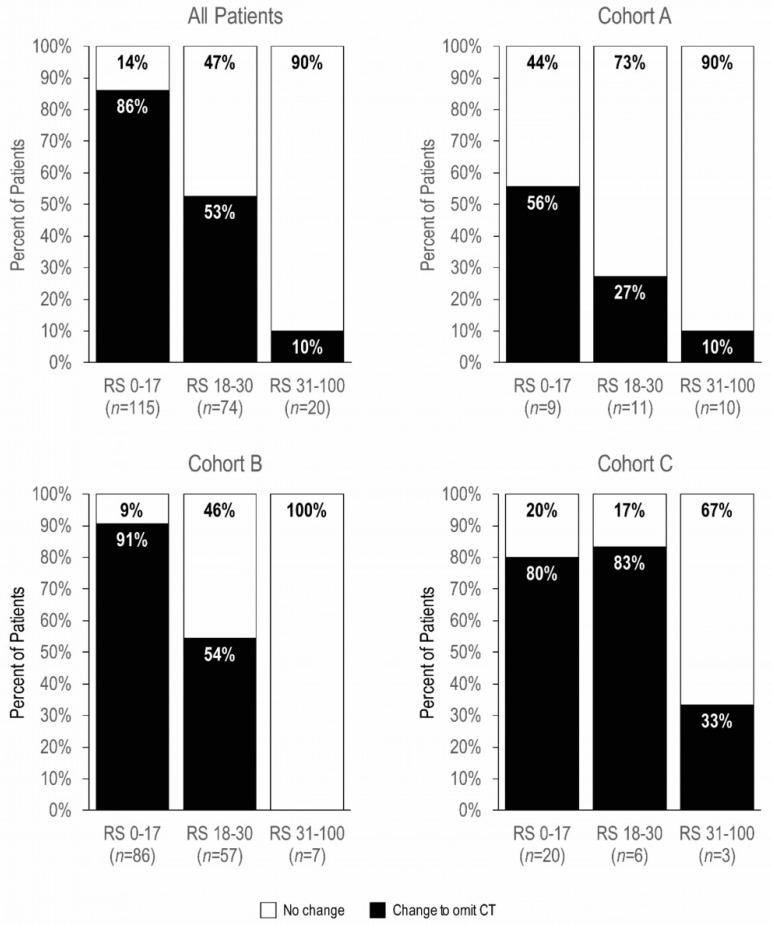
Change in treatment recommendations after 21-gene testing, by cohort and Recurrence Score group (*n* = 209). Change to omit CT indicates a change from CT + ET to ET in Cohorts A and B, and from CT to ET in Cohort C. CT, chemotherapy; CT + ET, chemotherapy plus hormonal therapy; ET, hormonal therapy alone; RS, Recurrence Score result.

**Figure 5 cancers-15-01529-f005:**
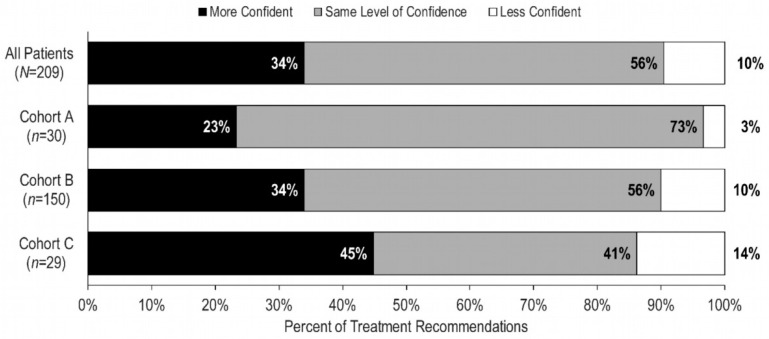
Physicians’ levels of confidence in post-test treatment recommendations for patients with Recurrence Score results (*n* = 209).

**Table 1 cancers-15-01529-t001:** Patient and disease characteristics, by cohort (*n* = 219).

		All Patients (*n* = 219)	Cohort A N0 (*n* = 30)	Cohort B N+ (*n* = 158)	Cohort C Neoadjuvant (*n* = 31)
Age, years	Median (IQR)	55 (47–66)	59 (51–67)	55 (47–66)	49 (44–59)
	≤50	78 (36%)	7 (23%)	54 (34%)	17 (55%)
	>50	141 (64%)	23 (77%)	104 (66%)	14 (45%)
Nodal status	N0/N1mi	59 (27%)	30 (100%)	0	29 (94%)
	N+	160 (73%)	0	158 (100%)	2 (6%)
Tumor grade	I	56 (26%)	0	47 (30%)	9 (29%)
	II	127 (58%)	0	111 (70%)	16 (52%)
	III	31 (14%)	30 (100%)	0	1 (3%)
	Unknown	5 (2%)	0	0	5 (16%)
Tumor classification	T1a	2 (1%)	0	2 (1%)	0
	T1b	20 (9%)	4 (13%)	16 (10%)	0
	T1c	111 (51%)	17 (57%)	88 (56%)	6 (19%)
	T2	79 (36%)	8 (27%)	50 (32%)	21 (68%)
	T3	7 (3%)	1 (3%)	2 (1%)	4 (13%)
Recurrence Score	Median (IQR)	17 (12–22)	22 (17–40)	17 (12–20)	16 (13–22)

IQR, interquartile range; N0, node-negative; N1mi, micrometastases; N+, node-positive (1–3 nodes).

## Data Availability

Data presented in this study are available on request from the corresponding author.

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
