# Peer review of "Impact of the 21-Gene Assay in Patients with High-Clinical Risk ER-Positive and HER2-Negative Early Breast Cancer: Results of the KARMA Dx Study"

_cancers, 2023, doi:10.3390/cancers15051529_

Round 1
Reviewer 1 Report
The manuscript titled“Impact of the 21-gene Assay on Treatment Decisions in Patients 2 with High-Clinical Risk Estrogen Receptor-positive and HER2- 3 negative Early Breast Cancer: Results of the KARMA Dx Study” by Llombart-Cussac et al., is well communicated.
1. Did the authors study any other biomarker (except Ki67)
2. Were there any adverse effects associated with the study? Those could be tabulated
3. The title needs to be precise.
Author Response
Dear Editor
The reviewers' comments are developed in the attached file.
Sincerely
Dr. LLombart Cussac

Reviewer 2 Report
The authors assessed the impact of the 21-gene test on the treatment decisions in real-world practice for a cohort of patients with clinically high-risk ER+, HER2− EBC for whom CT was indicated without testing and clinicians had doubts about the real benefit of CT. They aimed to define the utility of the 21-gene test in three cohorts of patients with intermediate- to high-risk clinical or pathological characteristics that excluded them from local access to the 21-gene test and for whom local or national guidelines (in Spain) were not conclusive regarding the use of (neo)adjuvant CT. They show that granting access to the 21-gene testing led to clinicians to reject CT in 67% of patients with well-defined intermediate- to high-risk clinicopathological ER+, HER2− EBC.
English language and style are fine and the article is easy to read. Study design and methods are described in enough details. Authors understand and clearly indicate limitations of the study. The paper may be accepted in present form.
Author Response

(The authors gave the same response as above.)

Reviewer 3 Report
The authors report that the Oncotype DX assay has made it possible to redetermine the subtype of ER-positive and HER2-negative breast cancer patients. If this assay method is used clinically, cancer patients will benefit because it will reduce the likelihood of inappropriate treatment.
comments:
1. The names of the 21 genes are given in reference 19, however, I think it would be better to show them in a table or something in this paper as well.
2. Heat map listing the expression levels and scores of the genes used in the assay would contribute to the reader's understanding.
Author Response

(The authors gave the same response as above.)

Round 2
Reviewer 1 Report
The authors have modified the manuscript as per comments.